# To Beam Or Not To Beam:
# That is a Question of Cooperation
# for Language GANs

**Thomas Scialom**[*‡], **Paul-Alexis Dray**[*], **Sylvain Lamprier**[‡],
**Benjamin Piwowarski**[◇‡], **Jacopo Staiano**[*]
[◇] CNRS, France
[‡] Sorbonne Université, CNRS, LIP6, F-75005 Paris, France
[*] reciTAL, Paris, France
thomas@recital.ai

## Abstract

Due to the discrete nature of words, language GANs require to be optimized from rewards provided by discriminator networks, via reinforcement learning methods. This is a much harder setting than for continuous tasks, which enjoy gradient flows from discriminators to generators, usually leading to dramatic learning instabilities. However, we claim that this can be solved by making discriminator and generator networks cooperate to produce output sequences during training. These cooperative outputs, inherently built to obtain higher discrimination scores, not only provide denser rewards for training, but also form a more compact artificial set for discriminator training, hence improving its accuracy and stability. In this paper, we show that our *SelfGAN* framework, built on this cooperative principle, outperforms Teacher Forcing and obtains state-of-the-art results on two challenging tasks, Summarization and Question Generation.

## 1 Introduction

Natural Language Generation encompasses tasks such as Machine Translation, Summarization or Data To Text generation. The real life applications are numerous, but require highly reliable and fluent models. Despite significant advances, state-of-the-art models are still known to be *de*-generated, with outputs containing repetitions and even nonfactual information i.e. hallucination [13].

Among the culprits is a limitation of Teacher Forcing [37]: the loss is computed at a token level while the aim is to produce complete sequences. Moreover, while a single ground-truth reference is considered correct, several realizations of the same content may exist. Finally, the model is subject to Exposure Bias [27], i.e. a mismatch between training and inference distributions – in the latter, the model has no access to ground truth for the previously generated tokens. The literature has considered this mismatch responsible for the lower quality observed when generating longer sequences [2, 16].

To overcome such Teacher Forcing limitations, a consensus has emerged: a sequence level objective should be introduced [27, 41]. A body of work has proposed to use Reinforcement Learning (RL) with standard NLG metrics like BLEU [39] or ROUGE [23]. However, NLG metrics are known to not reflect well human judgement [21], which explains why the resulting models tend to be qualitatively worse than their MLE baselines [3]. To move toward less biased metrics, a natural alternative is to evaluate the output with a learned discriminator. An ideal discriminator would not be biased w.r.t. to its training set, and could therefore be considered as a perfect metric that matches human consensus. Note that discriminators are already reported to be highly accurate to distinguish human written texts from machine generated ones [31, 43].

35th Conference on Neural Information Processing Systems (NeurIPS 2021).

In light of this observation, two concurrent approaches have been explored: i) at training time, using *Generative Adversarial Networks* [42]; and ii) at inference time, via cooperative decoding [11]: a discriminator guides the search algorithm, such that *the generator and the discriminator cooperate* to select the generated tokens. These approaches pursue the same objective: producing texts more similar to what a human writes.

Both methodologies suffer from specific limitations. Cooperative decoding algorithms rely on a discriminator that re-ranks a *limited* set of candidates selected by the generator.[1] Hence, cooperative decoding algorithms are limited by the generator ability to rank relevant tokens in a good enough position. On the other hand, language GANs are learned via reinforcement learning due to the discrete nature of text. This makes them particularly unstable to train, and usually fall short compared to Teacher Forcing [3]. In standard Language GANs, the discriminator provides a reward for the entire sequence, which can be difficult to exploit by the generator due to its sparsity [6].

In this paper, we propose *SelfGAN*, a framework to learn language GANs in a *Self*-training process where the signal from the discriminator is passed to the generator in a completely new way. We consider cooperative algorithms as a way to infuse the discriminator signal. We start from a simple observation: outputs obtained via cooperative decoding are more human-like, compared to their generator-only counterparts. Inspired by recent knowledge distillation approaches, we propose to consider *cooperative outputs* as targets in a Teacher Forcing training process: cooperative decoding stands as a teacher we attempt to imitate through the generator network. Just like a standard GAN, both the generator and the discriminator are trained at each step. While the generator improves, it becomes adversarial to the discriminator, which benefits from the cooperative generation. The discriminator, now trained on improved sequences, also contributes to improve the cooperative generation, and so forth. Note that in *SelfGAN*s the discriminator is only used to drive the cooperative generation and never to provide a reward signal like in standard Language GANs.

*SelfGAN* can be implemented with any cooperative decoding algorithm. Current cooperative approaches [7, 31] rely on "myopic" algorithms like Beam Search or Sampling that generate the tokens left-to-right. The model has to always predict the next word, and can never look back and revise past choices. In some cases, despite all the candidates being judged to likely not be human by the discriminator, the model is locked in a dead-end. This behavior is quite unnatural for humans – who often proofread their texts. We refer to this phenomenon as the *left-to-right curse*.

To address this *left-to-right curse*, we introduce *Coop-MCTS*, a new decoding algorithm based on Monte Carlo Tree Search (MCTS) [5, 14]. We compare *Coop-MCTS* to state-of-the-art cooperative decoding algorithms in two scenarios: i) inference time, as the decoding algorithm; and ii) during training, as the cooperative algorithm in *SelfGAN*. In both scenarios, we show that the respective resulting outputs are more likely to look like human texts and improve all the automatic metrics.

All in all, our contributions can be summarized as follows:

1. **SelfGAN** We propose a new training framework based on cooperative decoding, wherein the generated sequences are used as ground truth;

2. **Coop-MCTS** We improve cooperative decoding with a new decoding algorithm, *Coop-MCTS*, offering a solution to the left-to-right limitation of current search methods;

3. We show that combining *SelfGAN* and *Coop-MCTS* compare favorably to prior state-of-the-art results on two challenging tasks, Summarization and Question Generation.

## 2   Related Work

**Beyond Teacher Forcing** To mitigate the limitations inherent to Teacher Forcing, various alternatives have been proposed. In Scheduled Sampling, Bengio et al. [2] proposed to condition the generation not only on the ground truth tokens, but also the generated ones. Given that only one reference is available, this introduces a new mismatch when computing the loss, this time between the generated tokens used to condition the model, and the target tokens. To take into account multiple possible

---

[1]Opposed to a generator that outputs probabilities for the entire vocabulary $V$ at once, a discriminator outputs the likelihood for a specific sequence to be human-written or machine-generated. Calculating the discriminator probability for every possible sequence is therefore not realistic, as the computation grows exponentially at a pace of $V^l$ where $V$ is the vocabulary size and $l$ the sequence length.

references, using a sequence level metric is a potential alternative. In Mixer, Ranzato et al. [27] chose BLEU, the standard metric to evaluate Machine Translation. Since it is not differentiable, the task is framed in a Reinforcement Learning setup where the reward corresponds to the BLEU score given a sampled sequence of tokens. Paulus et al. [23] applied the same method to Summarization, using this time ROUGE. While the results improved in terms of ROUGE, the human evaluation found that the generated summaries were rated worse in term of fluency than the MLE baseline. The model learns to take advantage of metric biases, while being less correct according to human judgement.

**Language GANs** In theory, a perfect discriminator would be able to judge if an output corresponds to the data distribution or not. Discriminators could therefore be an interesting alternative reward compared to other metrics. In practice, we need to train the discriminator jointly with the generator, framing the task as a GAN. Language GANs are known to underperform MLE [3], due to the unavoidable sparsity of a discriminator reward. A large body of works have proposed denser rewards: ranking or comparative discriminators [4, 17, 45], a sequential discriminator where the rewards are provided at each time step of the generation [35, 6]. More recently, Scialom et al. [30] proposed to stabilize the GAN training by lowering the Softmax temperature to explore more structured outputs, and closer to the generator distribution.

In this work, our proposed framework allows to propagate the discriminator signal in a cooperative way, which can be seen as an alternative solution to the sparsity of the reward and the training stability.

**Knowledge Distillation** *SelfGAN* has a connection with knowledge distillation [12], where a student is trained on outputs from the teacher. In particular, self distillation using only a generator has shown to improve generalisation on image GANs [44] by acting as label smoothing. To the best of our knowledge, this work is the first to propose the idea of augmenting the teacher by coupling a discriminator to a generator. Beyond GANs, *SelfGAN* could serve for other applications using distillation, e.g. in semi-supervised methods that use a teacher model to create synthetic labels for unlabeled examples [33, 40].

**Monte Carlo Tree Search in NLG** Despite important successes in games [29, 36], very few works have attempted to apply MCTS to NLG. Kumagai et al. [15] proposed to employ context-free grammar rules combined with a n-gram language model and explore the space of grammatically correct texts via a MCTS. In the context of commercial e-commerce agents, Mukherjee [19] proposed to optimise with a MCTS a scoring function designed to reward grammatical correctness.

## 3 SelfGAN

---
**Algorithm 1** *SelfGAN*

---
1: **Input:** a generator $gen$, a discriminator $discr$, and a cooperative decoding method $decod_{coop}$
2: **for** n epochs **do**
3:     **for** $X$, $S_{ref}$ in training set **do**                                    ▷ *Start Training*
4:         $S_{coop} \leftarrow decod_{coop}(X, gen, discr)$
5:         $gen$.train(srcs=$X$, tgts=$S_{coop}$)    ▷ *Standard maximum likelihood but with $S_{coop}$ as the*
    *target, and not $S_{ref}$*
6:         $discr$.train(srcs=$X$, human_exs= $S_{ref}$, machine_exs=$S_{coop}$)

---

The difficulty of GAN-based approaches for NLP tasks lies in the fact that no gradient flow can be propagated from the discriminator to the generator. As discussed above, approaches from the literature circumvent this difficulty by employing RL approaches, using discriminator scores as rewards to train the generator. However, such approaches induce great instabilities in the learning process, due to the use of a non-stationary reward function in addition to the high variance associated to monte-carlo estimations of RL.

The idea in our *SelfGAN* approach is to transfer the sparse signal of the discriminator, classically used as rewards for a RL procedure, to the sampling mechanism of sequences that have to be favored through MLE. In that way, *SelfGAN* starts from a pretrained generator, that we fine-tune using sequences $S_{coop}$ provided by a cooperative decoding process $decod_{coop}$ for each condition in the training set $X$. This process, detailed in the next section, uses both the generator and a discriminator network to output human-like sequences $S_{coop}$, for which we improve the generator likelihood via classical maximization: $\max_\theta \sum_{(x,s) \in (X, S_{coop})} \pi_\theta(s|x)$, where $\pi_\theta(s|x) = \prod_{t=1}^{|s|} \pi_\theta(s_t|s_{0:t-1}, x)$

stands for the generator probability of sequence $s$ given the conditioning input $x$, with $\pi_\theta$ implemented as a neural architecture with a softmax output function.

At each iteration of the training procedure, the discriminator network is optimized as a binary classifier on i) the human references and ii) the machine generated via the cooperative sequences:

$$\frac{1}{|H|} \sum_{(x,s_{ref}) \in H} \log(D(x, s_{ref})) + \frac{1}{|G|} \sum_{(x,s_{coop}) \in G} \log(1 - D(x, s_{coop}))$$

where $x$ is the source input, $H$ is the set of pairs associating $x$ with a human written text $s_{ref}$ from the data distribution, and $G$ is a set of pairs with generated outputs $s_{coop}$. $D(x, s)$ stands for the probability, provided by the discriminator network, that sequence $s$ is a human reference for condition $x$. In order to effectively guide the cooperative process at each step, the discriminator needs to be sequential: consistently with [31], we use a left-to-right mask during training, allowing discriminator predictions for unfinished sequences.

Please note that, by construction of the cooperative decoding process, we have with high probability at each iteration $D(x, s_{coop}) >= D(x, s_{gen})$ for any condition $x \in X$, with $s_{coop}$ a cooperative decoded sequence for $x$ and $s_{gen}$ a sequence directly sampled from the generator according to $\pi_\theta(s|x)$. Based on this observation, and provided that the discriminator is sufficiently trained at each step, the generator is trained such that the probability of predicting human-like sequences is maximized. This process i) allows us to consider a sequence level metric, and ii) offers more stability compared to Reinforcement Learning, as we observe in our experiments (see section 6). Note also that, contrary to RL approaches which have to find a good balance between discriminator and generator capacities, our approach does not suffer from Vanishing Gradient [1], since discrimination is only used for decoding, in a cooperative process for generator training. We depict the *SelfGAN* in Algorithm 1.

## 4 Decoding Mechanisms

### 4.1 Standard Practices: Generator-only

At decoding time, two different approaches are commonly used in NLG: Sampling and Beam Search. They respectively correspond to two different objectives.

**Sampling** To obtain diverse outputs, it is common to sample tokens from the model distribution. In particular, this is mandatory when there is no input at all, i.e. *Unconditional NLG*, for instance GPT [24]. However, the longer the sequence, the more likely to sample a token from the tail of the distribution, causing degeneration [13]. To mitigate this issue, common practices are to lower the Softmax Temperature and keeping only the Top K tokens [10] / the Top P probability mass [13].

**Beam Search** is the standard algorithm to approximate the sequence maximising the output probability, by maintaining K candidates at each step. Its usage suits better *conditional* NLG tasks, where the diversity arises from the variety of conditioners inputs, e.g. in Summarization.

### 4.2 Cooperative Decoding: Combining a Discriminator and a Generator

Subject to exposure bias, neither Sampling or Beam Search are satisfying: the outputs produced are easily identified by a discriminator [31], indicating that they differ from human written text. In light of this, two concurrent works have recently proposed to use the discriminator during the decoding.

**DAS$_{local}$ - Reranking Step By Step** In Discriminative Adversarial Search [31] a discriminator re-ranks the sub-sequence candidates *at each decoding step* of a Beam Search, in order to favor human-like outputs.

**DAS$_{global}$ - Reranking Complete Sequences** In a concurrent work [7] a very similar cooperative method is proposed: this time, $N$ complete sequences are sampled from the auto-regressive model. The $N$ sequences are scored by a discriminator, allowing to select the one with the highest probability to be human-like. Since the discriminator re-ranking is computed on a complete sequence, we refer to this method as DAS$_{global}$, as opposed to DAS$_{local}$.

### 4.3 Coop-MCTS: Cooperative Decoding beyond the *Left-To-Right Curse*

It can happen that all sequence candidates are judged by the discriminator to be machine-like rather than human-like. In such case, the cooperative decoding is stuck in a *dead end*; such limitation is unsatisfactory. Neither $DAS_{local}$ or $DAS_{global}$ have the ability to revise their previous decisions.

To cope with those limitations of myopic decoding strategies, we propose to consider an adaptation of MCTS for NLG. Just like in the context of games [36], we consider a policy network $\pi$, the generator, that outputs a probability over all the possible actions (tokens) at each step of the sequence. The discriminator $D$ corresponds to the value network. In MCTS, the trajectories are explored to build a tree following three steps:

1. **Selection** starting from the root, children nodes tokens $\omega$ are selected among the vocabulary $\mathcal{V}$ recursively w.r.t. the PUCT algorithm [28, 36]:

$$\omega = \arg\max_{\omega \in \mathcal{V}} \left( Q(s, \omega) + c_{puct} \pi_\tau(\omega \mid s) \frac{\sqrt{\sum_b N(s, b)}}{1 + N(s, \omega)} \right) \tag{1}$$

   where $Q$ is the value of taking action $\omega$ in state $s$: in NLG, this corresponds to selecting a token among the vocabulary at step $i$ given the source context and the sub-sequence $\omega_0, ..., \omega_{i-1}$. $c_{puct}$ is a constant, $\tau$ the temperature that scales the Softmax, and $N(s, \omega)$ the number of times the token $\omega$ has been chosen in state $s$. We stop the loop when a node $s_o$ has not been expanded yet, i.e. the discriminator $D$ has not calculated its value.

2. **Extension** Given the selected node, we calculate the distribution probability from the generator $\pi(\omega \mid s_o)$. We apply nucleus sampling [13] to filter out the less likely tokens and reduce the number of actions. The remaining tokens constitute the children nodes, associated to their corresponding probability. At the same time, we calculate the value of the current state $D(s_o)$ that allows to compute the backup step.

3. **Backup** we update $Q$ for all the nodes that led to $s_o$ such that $Q \leftarrow \max\left(Q, D\left(s_o\right)\right)$. Note that we choose to use the max instead of the average for the following reason: the value network, i.e. a discriminator, becomes more accurate as the candidate sequence grows (see Figure 2 in [31]), hence if a long sequence is judged human by the discriminator, any of its sub-sequences should be considered human-like as well. In contrast, a long sequence can be machine-like despite starting in a very human-like manner: the beginning sub-sequence should keep its human-like score.

These three steps are computed for a restricted number of simulations. Then, the next token corresponds to the root child with the most visit counts. The process continues step by step to generate the next token, until reaching either the special token End Of Sentence, or the maximum length.

## 5 Experimental Details

### 5.1 Datasets

To measure the effectiveness of *SelfGAN*, we experiment on two standard conditional NLG tasks: Question Generation (QG) and Summarization, consistently with previous works [8, 30]:

- **Question Generation:** we used the SQuAD dataset [26], consisting of 100K triplets of Wikipedia paragraphs, factual questions, and their answers.
- **Summarization:** we used the CNN/Daily Mail dataset (CNNDM) [20], consisting of 300K news articles, paired with their corresponding summaries. The summaries are formed of multiple sentences, making the amount of tokens to generate much larger than for Question Generation.

### 5.2 Models Reported

**MLE** the first baseline we consider is a standard model trained via teacher forcing. As for all our experiments, we initialised the seq2seq with T5 [25], as detailed in Section 5.4.

**ColdGAN** we consider as a second baseline the current state-of-the art for language GANs, ColdGAN [30]. The authors proposed to lower the temperature when sampling the sequences during training, with the objective of stabilizing the training process.

**SelfGAN** can be based on any cooperative decoding algorithm. To train *SelfGAN*, we therefore experiment the three different cooperative algorithms described in Section 4 (DAS$_{Local}$, DAS$_{Global}$, and *Coop-MCTS*) and report the results for the corresponding *SelfGAN*: *SelfGAN$_{DAS-Local}$*, *SelfGAN$_{DAS-Global}$*, and *SelfGAN$_{Coop-MCTS}$*.

**Decoding Method at inference time** For each model, any decoding method can be applied at inference time, independently from the training scheme. Therefore, for all the models described above, we report the results given each decoding method previously described (Section 4): Beam Search, DAS$_{Local}$, DAS$_{Global}$, and *Coop-MCTS*.

To the best of our knowledge, GANs and Cooperative decoding have never been directly compared before this work. A fortiori, this is the first time that a GAN model is tested with a Cooperative decoding method at inference. We investigate possible distillation effects in Section 6.

### 5.3 Metrics

To compare the different models, we report two type of metrics: *n-gram based* and *discriminator*.

**N-gram based** We report the standard BLEU [22] and ROUGE [18]. Both measure an overlap of n-grams between the reference and the evaluated text. They differ in that BLEU is precision oriented while ROUGE is rather recall oriented.

**Discriminators** Both BLEU and ROUGE suffer from the aforementioned limitations. We therefore propose to consider discriminators for model evaluation. Intuitively, they measure how model outputs are similar to what a human would have written. We consider two different discriminators:

- **Base** is a discriminator trained on the MLE baseline outputs generated via beam search. It allows to measure the corresponding improvement from the MLE baseline. Note that it corresponds to the initial discriminator in all the GANs experiments, and the discriminator used in the cooperative search for the MLE baseline.

- **Base+** Since the *Base* discriminator plays a role in all our experiments (except MLE+Beam Search), it is possible that a model that makes use of this *Base* obtains better *Base* results, despite bringing new biases and *de*-generation behaviors. For this reason, we also report *Base+*, a discriminator fine-tuned on all the different model outputs together. *Base+* is never used by any model at training or inference time. It is thus more robust toward an undesirable adversarial generation mode, while still being comparable for the different experiments. We argue that a higher *Base+* score indicates a real improvement beyond potential bias.

### 5.4 Implementation Details

For all experiments, we used the T5-small [25] architecture.[2] Using 4 Nvidia V100 SXM2 GPUs, *SelfGAN*$_{Coop-MCTS}$ training and evaluation takes respectively 26 hours and 1 hour on CNN/DM; 6 and 0.5 hours on SQuAD. Compared to 2 (0.5) hours to train via MLE on CNN/DM (SQuAD), we identify in the computational cost the main limitation of our work.

## 6 Results and discussion

### 6.1 Conditional Text Generation

In Table 1, we report the results for all the previously trained generators, with the different decoding algorithms presented in Section 4. By 'model', in the following, we refer to the couple composed by a trained generator and a decoding algorithm.

We report BLEU4, ROUGE-1, ROUGE-L along with scores for the discriminators *Base* and *Base+*, computed as the percentage of outputs considered as human by a given discriminator model. *Base* was only trained on *MLE+Beam Search* outputs. As expected, by further training on the outputs generated by all the different models, *Base+* has a higher accuracy, which consistently results in lower scores compared to *Base*.

---

[2]As implemented in HuggingFace transformers [38].

| Generator | Question Generation | | | | | Summarization | | | | |
|---|---|---|---|---|---|---|---|---|---|---|
| Decoder | **B4** | **R1** | **RL** | **Base** | **Base+** | **B4** | **R1** | **RL** | **Base** | **Base+** |
| **MLE** | | | | | | | | | | |
| BeamSearch [25] | 19,7 | 45,2 | 41,1 | 15% | 15% | 15,9 | 42,3 | 40,4 | 9% | 8% |
| DAS$_{local}$ [31] | 19,9 | 45,2 | 41,1 | 28% | 19% | 16,6 | 43,8 | 40,9 | 17% | 11% |
| DAS$_{global}$ [7] | 20,0 | 45,2 | 41,2 | 20% | 17% | 16,2 | 44,1 | 41,9 | 12% | 9% |
| Coop-MCTS | 19.8 | 45,3 | 41,5 | 33% | 21% | 16,3 | 42,5 | 40,6 | 20% | 12% |
| **ColdGAN** | | | | | | | | | | |
| BeamSearch [30] | 19.9 | 45,2 | 41,4 | 26% | 17.9% | 16,3 | 42,8 | 40,7 | 15% | 10% |
| DAS$_{local}$ | 19.8 | 45,3 | 41,1 | 31% | 20% | 15,9 | 42,5 | 42,0 | 19% | 11% |
| DAS$_{global}$ | 20,2 | 45,6 | 41,5 | 26% | 18% | 16,6 | 44,6 | 41,2 | 16% | 10% |
| Coop-MCTS | 19,9 | 45,4 | 41,2 | 39% | 22% | 15,9 | 44,2 | 41,2 | 23% | 12% |
| **SelfGAN$_{DAS_{loc}}$** | | | | | | | | | | |
| BeamSearch | 20,2 | 45,4 | 41,6 | 27% | 21% | 16,9 | 44,2 | 42,5 | 16% | 11% |
| DAS$_{local}$ | 20,5 | 45,5 | 41,7 | 30% | 23% | 16,9 | 44,4 | 41,9 | 18% | 13% |
| DAS$_{global}$ | 20,1 | 45,4 | 41,7 | 33% | 20% | 16,6 | 44,0 | 42,3 | 19% | 11% |
| Coop-MCTS | 20,4 | 45,5 | 41,8 | 39% | 23% | 16,4 | 43,8 | 42,8 | 23% | 13% |
| **SelfGAN$_{DAS_{glob}}$** | | | | | | | | | | |
| BeamSearch | 20,4 | 45,5 | 41,7 | 24% | 19% | 16,9 | 43,0 | 41,5 | 14% | 11% |
| DAS$_{local}$ | 19,9 | 45,4 | 41,3 | 32% | 22% | 15,9 | 42,7 | 40,6 | 18% | 12% |
| DAS$_{global}$ | 20,7 | 45,6 | 41,9 | 29% | 20% | 17,0 | 43,7 | 42,6 | 17% | 11% |
| Coop-MCTS | 20,0 | 45,3 | 41,4 | 40% | 24% | 16,1 | 43,4 | 42,3 | 23% | 13% |
| **SelfGAN$_{Coop-MCTS}$** | | | | | | | | | | |
| BeamSearch | 20,5 | 46,6 | 42,6 | 34% | 21% | 17,0 | 42,8 | 41,5 | 20% | 13% |
| DAS$_{local}$ | 20,6 | 46,7 | 41,7 | 42% | 24% | 16,6 | 43,7 | 42,8 | 25% | 13% |
| DAS$_{global}$ | 20,5 | 46,6 | 41,7 | 39% | 21% | 16,5 | 42,8 | 40,9 | 23% | 12% |
| Coop-MCTS | **21,1** | **48,9** | **44,7** | 40% | **26%** | **17,5** | **43,5** | **42,3** | 23% | **15%** |

Table 1: Results of our experiments on QG (left) and Summarization (right). For each generator, we report the results with the four different decoders. The reported metrics correspond to BLEU4 (B4), ROUGE-1 (R1), ROUGE-L (RL) and the discriminators Base and Base+ as described in Section 5.3. For Base and Base+ the scores correspond to the probability of being human, so higher is better for all the metrics. For *SelfGAN*$_{MCTS}$, we experimented with 5 different seeds and the standard deviation is always inferior to 0.1 for BLEU4 and ROUGE, and inferior to 0.5% for Base and Base+.

We start by focusing on the MLE results to compare the different decoding mechanisms. We observe that all the cooperative searches outperform Beam Search. Regarding Base and Base+ metrics, DAS$_{Local}$ compares favorably to DAS$_{Global}$. We hypothesize that invoking the discriminator to rank at each step can have more impact than using it only once on fully decoded sequences. Finally, our proposed *Coop-MCTS* obtains the best results by a large margin.

Regarding the different GANs, we first compare them given the default decoding mechanism, i.e. Beam Search. The three versions of *SelfGAN* compare favorably to MLE and ColdGAN on both n-gram based metrics and discriminators metrics. Among *SelfGAN*s, *SelfGAN*$_{Coop-MCTS}$ obtains the best results: given a Beam Search decoding, it obtains the best BLEU, ROUGE-1 and ROUGE-L on the two tasks (respectively 17.2; 44.3; 40.6 on QG and 12.3; 38.6; 36.7 on Summarization). The performance in term of Base and Base+ for *SelfGAN*$_{Coop-MCTS}$ is even more important in comparison to the other models (34.1%; 21.9% on QG and 20.2%; 12.7% on Summarization).

Both GAN at training time and Cooperative decoding at inference time pursue the same objective: to obtain better outputs that look like human texts. Would a generator trained via GAN, coupled with a Cooperative Decoding mechanism for inference result into a cumulative improvement from the two methods? First, on both ColdGAN and three *SelfGAN*s, we can observe that adding a Cooperative Decoding method allows to gain significant improvement on Base and Base+. In particular, it is interesting to note that for *SelfGAN* an additional pattern seems to emerge: using the same cooperative decoding algorithm both during training and inference seems to provide additional gains. The best performance is achieved with the generator *SelfGAN*$_{Coop-MCTS}$ paired with the decoding *Coop-MCTS*. Compared to MLE via Beam Search, it obtains a final improvement superior to 1 point in term of ROUGE and BLEU. The relative improvement for Base+ is significant: from 15.2% to 26.2% on

| Model | T=0.5 | T=1 | T=2 |
|---|---|---|---|
| MLE+Sample | 0.42;0.29 | 0.31;0.11 | 0.18;0.07 |
| ColdGAN+Sample | 0.47;0.21 | 0.33;0.08 | 0.22;0.06 |
| MLE+Coop-MCTS | 0.45;0.22 | 0.34;0.10 | 0.21;0.06 |
| SelfGANCoop-MCTS+Coop-MCTS | 0.48;0.20 | 0.37;0.09 | 0.24;0.05 |

Table 2: Results on Unconditional Text Generation for samples realized at three different temperatures, in terms of BLEU Vs Self-BLEU (higher better;lower better).

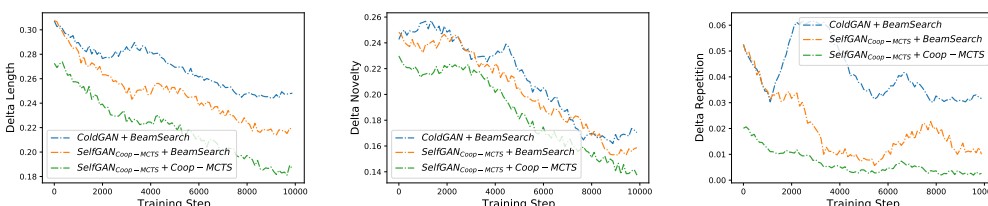

Figure 1: Average difference for Summarization between human references and model outputs for the Length (Left), the Novelty (Middle), and the 3-grams repetitions (Right) during training. The closer to 0 the less differences w.r.t. gold-references.

QG and from 8.6% to 15.3% on Summarization. This corresponds to almost twice more outputs that sound human according to the Discriminator metric.

## 6.2 Unconditional Text Generation

We follow the ColdGAN setup: we compared our proposed approaches on the EMNLP2017 News dataset. The evaluation takes into account both the quality and the diversity. Consistently with previous works (e.g. ColdGAN, ScratchGAN, LeakGAN), we use the following metrics:i) BLEU-5 for measuring the quality (higher better); ii) Self-BLEU-5 for measuring the diversity (lower better).

To obtain a finer comparison between models, Caccia et al. [3] proposed to draw the curve of BLEU vs self-BLEU, by sampling with various temperatures at inference.

We denote as the standard method to generate text in this setup, as used in all the previous works we are comparing to. It is a simple left to right decoding where, at each step, a token is sampled among the Softmax probabilities scaled by the temperature.

In our Coop-MCTS, the probability of a token is given by its visit counts during the simulations. In Conditional generation, we select at each step the token with the maximum number of counts. In Unconditional generation, we sample from the tokens counts distribution.

Overall, the results are consistent with the experiments on Conditional Generation: the MLE generator decoded with our proposed MCTS (3rd row) obtains:

1. significantly better slightly lower results than ColdGAN decoded with Sample(2nd row)

2. results than when the same MLE decoded with Sample(1st row);

3. SelfGAN decoded with MCTS (4th row) obtains the best results.

## 6.3 Discussion

**Human-like features during training** In NLG, various rules are often integrated into the Beam Search to improve the quality of the outputs, for instance a length penalty [34] or an interdiction for 3-grams repetitions [23, 8]. Such a need to hard code these rules indicates a discrepancy between the human output characteristics and what the model has learned. In particular, Scialom et al. [31] reported the difference between DAS and the human reference for: i) **Length**: the average number of tokens per output; ii) **Novelty**: percentage of tokens in the output that were not present in the source text; iii) **N-gram repetition**: percentage of N-grams that occur more than once in the output. To measure how *SelfGAN* learns these features by itself, we report in Figure 1 the evolution of these

| Generator | Decoder | Consistency | Coherence | Fluency | Relevance |
|---|---|---|---|---|---|
| MLE | BeamSearch | 3.9 | 3.1 | 4.1 | 3.2 |
| MLE | Coop-MCTS | 3.4** | 3.5** | 3.8 | 3.6** |
| ColdGan | BeamSearch | 3.8 | 3.3 | 4.2 | 3.5 |
| ColdGan | Coop-MCTS | 3.4** | 3.6** | 4.0 | 3.7** |
| SelfGAN$_{\text{Coop-MCTS}}$ | BeamSearch | **4.0** | 3.5** | **4.3*** | 3.9** |
| SelfGAN$_{\text{Coop-MCTS}}$ | Coop-MCTS | 3.9 | **3.9**** | 4.0 | **4.2*** |

Table 3: Human Evaluation on Summarization. Two tailed t-test results are reported for each model compared to MLE+BeamSearch (*: $p < .01$, **: $p < .001$).

statistics during training: we observe that *SelfGAN* constantly reach statistics more similar to human references than ColdGAN.

**Human Evaluation** We conduct a human evaluation to measure the models performances beyond automatic metrics. We limit the evaluation to three generators (MLE, ColdGAN, and *SelfGAN*$_{\text{Coop-MCTS}}$) and two decoding methods (Beam Search and *Coop-MCTS*), for a total of 6 different models. Three professional English speakers rated 300 sampled summaries and followed the same protocol from Fabbri et al. [9]. Four dimensions are evaluated on a Likert scale from 1 to 5 (the higher the better):

1. **Consistency:** the proportion of facts in the summary correct w.r.t. the source text;
2. **Coherence:** how well-structured and well-organized is the summary;
3. **Fluency:** how fluent the summary is to read;
4. **Relevance:** the ratio between important and excess information in the summary.

From Table 3 we observe significantly better results for *SelfGAN*$_{\text{Coop-MCTS}}$ w.r.t. both MLE and ColdGAN. While *Coop-MCTS* decoding appears overall beneficial in terms of Coherence and Relevance, but scores lower on Consistency and Fluency, its combination with *SelfGAN*$_{\text{Coop-MCTS}}$ allows to obtain significant improvements on the former two dimensions while still maintaining comparable scores on the latter.

**Analysis** To further understand the benefits of selfGAN, we propose to analyze the evolution of the generator and discriminator networks through the learning process. In figure 2 (left), we first plot the average magnitude (L2 norm) of the discriminator gradients w.r.t. its parameters. We observe that $ColdGAN$ induces important instabilities for its discriminator over time, with a highly fluctuating gradient magnitude. Conversely, thanks to its cooperative decoding process, *SelfGAN* produces sequences that form a more compact set for discriminator training, a variance of gradient magnitude twice lower than $ColdGAN$, for a comparable magnitude in average. This discriminator stability is a first explanation for the improvements of the proposed approach.

In a second plot, given on the right of Figure 2, we report the collinearity of generator gradients for the generated samples from the model with those for the corresponding human references. Higher values indicate sampling strategies that induce a useful gradient flow for the generator. For ablation purposes, we first report values for a *"SelfGAN$_{BeamSearch}$"* approach, where we used a standard Beam Search to generate the training examples: note that it has no discriminator, hence it is not a GAN anymore. We can observe its divergence, as opposed to *SelfGAN*$_{\text{Coop-MCTS}}$, which emphasizes the importance of the cooperative decoding for producing the example used to train the model. For *SelfGAN*$_{\text{Coop-MCTS}}$ and $ColdGAN$, the gradients become more co-linear with human references through time, indicating a convergence of the process towards the human distribution. We observe that *SelfGAN*$_{\text{Coop-MCTS}}$ produces more useful sequences for achieving this convergence.

**Coop-MCTS as an alternative to the dead-end search** When analysing the behavior for the *Coop-MCTS* decoding, we observed in different examples that it provides an effective mean to revise generations that eventually ended up to be unlikely. To illustrate this, we report in Table 6.3 the different MCTS steps for an ambiguous example: the conditioned answer, *Super Bowl*, occurs at different places of the the input. Therefore, the model has to decide which specific mention of *Super Bowl* to focus on: at step 17, it considers its current generation as a dead end and decides to start on new node (*How*). The final output is a question that arguably sounds better than the initial one.

**Societal Impact** Reliable NLG models can have significant societal impact with beneficial applications such as efficient information access via automatic summarization or personalized student evaluation through question generation. Still, malicious actors can use the same technology to build

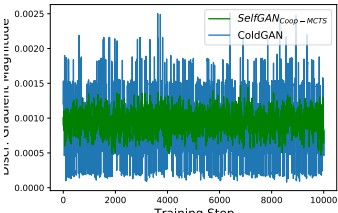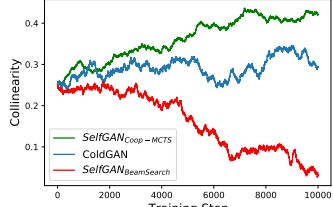

Figure 2: Left: Moving Average of the magnitude of the *discriminators* gradients during training. Right: collinearity of the *generators* gradients between the sampled texts and their corresponding human reference for *SelfGAN*Coop-MCTS, ColdGAN and *SelfGAN*BeamSearch. Both on Summarization.

**Conditioned Answer:** *Super Bowl*
**Context:**

Super Bowl 50 was an American football game to determine the champion of the National Football League (NFL) for the 2015 season. The American Football Conference (AFC) champion Denver Broncos defeated the National Football Conference (NFC) champion Carolina Panthers 24â€"10 to earn their third Super Bowl title. The game was played on February 7, 2016, at Levi's Stadium in the San Francisco Bay Area at Santa Clara, California. As this was the 50th Super Bowl, the league emphasized the "golden anniversary" with various gold-themed initiatives, as well as temporarily suspending the tradition of naming each Super Bowl game with Roman numerals (under which the game would have been known as "Super Bowl L"), so that the logo could prominently feature the Arabic numerals

**Step 01:** *What*
⋮
**Step 16:** *What was the name of the game that would have been known as "Super Bowl*
**Step 17:** *How*
⋮
**Step 46:** *How is called the American football game that determines the NFL champion?*

Table 4: Progressive results obtained by our *Coop-MCTS* decoding method on Question Generation during a simulation. Until the 16th step, the generation is left-to-right. Then, the cooperation mechanism kicks in, allowing the model to safely abort this beam, by restarting a new question with *How*. We report the cross-attention weights on the input context for step 16 (red) and 17 (blue).

tools detrimental to society, e.g. large scale creation of misleading (fake) news [24]. As argued by Zellers et al. [43], keeping this research open and under public scrutiny can be an effective defense.

# 7 Conclusion

In this paper we propose *SelfGAN*, a new framework to train Generative Adversarial Networks based on a cooperative decoding search. To overcome the left-to-right curse that limits standard search algorithms, we propose *Coop-MCTS*. We conducted extensive experiments on two challenging tasks: Summarization and Question Generation, obtaining state-of-the-art performance for *SelfGAN* both in terms of automatic metrics and within a human evaluation. As the stability of the discriminator looks to be crucial for language GANs, we plan for future works to still focus on increasing it through the definition of dynamic regularization mechanisms. Finally, we will explore how reference-less metrics, e.g. [32], can be combined to help the exploration during the decoding.

# 8 Acknowledgments

This work was partially performed using HPC resources from GENCI-IDRIS (Grant 2021-AD011012318).

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
