## A Appendix

### A.1 Implementation Details

In MCTS, sequence lengths are not aligned as in a standard left-to-right decoding algorithm. Therefore, we used a simple trick to enable efficient batching of sequences, that can be applied to any Language Model benefiting from a relative positional embedding [37]. We used a custom left padding that shifts the start of each sequences from a batch, so that all of their last tokens are aligned. In all our experiments, we used the T5-small [26] generator,[3] in which the embedding is relative.

For the discriminators, we frame the classification task as a text2text task where the model has to generate either the token *human* or *machine*. This allows to use again T5-small for all experiments, removing possible bias from architecture differences between the generator and the discriminator.

We start by training via Teacher Forcing a model corresponding to the MLE baseline. All our GANs are initialized from this MLE model. During training, we used a learning rate fixed to 5e-6 for both the discriminator and the generator, and a number of epochs set to 5.

We tested on a validation set different values for our hyper parameter $C_{puct} \in [1.0, 2.0, 3.0, 4.0]$ and found that 3.0 gives the best results. We thus only report the results with $C_{puct} = 3.0$. For the budget allocated to the MCTS we tested different number of simulations per token for the MLE model with ($n \in [5, 10, 25, 50, 100]$ and observed no significant improvement between 50 and 100. We hence used $n = 50$ for all our experiments.

We used 4 Nvidia V100 SXM2 GPUs for this project. SelfGAN$_{\text{Coop-MCTS}}$ training and evaluation takes respectively 26 hours and 1 hour on CNN/DM; 6 and 0.5 hours on Question Generation.

### A.2 Differences with [17]

In a concurrent work Leblond et al. [17] proposed MCTS as an alternative to Beam Search. There are two differences with our work.

First, the authors limit their study to MCTS as a decoding algorithm at inference time, and use a standard generator trained via MLE.

---

[3] As implemented in HuggingFace transformers [40].

Secondly, for the value network in the MCTS, they proposed to optimise a static metric, the BERTScore. However, we argue that these metrics are not reflecting human judgement [22], and models that maximise them are found to perform poorly [24]. Therefore, in their setup, improving BERTScore does not mean that the resulting model is better. Conversely, we chose a dyanmic metric, i.e. A discriminator, for our value network in our proposed Coop-MCTS, or any other cooperative decoding algorithm.

### A.3   Cooperation VS Competition

SelfGAN can be seen as an implicit solution for the reward sparsity problem, whereby the gradient from the reward is not tractable in language GANs. Conversely to prior works that have focused on denser rewards, in SelfGAN the sequence generation is directly driven by the discriminator to produce $s_{coop}$.

In addition, standard GANs are known to be particularly unstable for several reasons. In particular, a fine balance has to be found between the generator and the discriminator performance. If the discriminator becomes too strong compared to the generator, the reward is null, a phenomenon known as the Vanishing Gradient [1]. We emphasize that our proposed approach does not suffer from Vanishing Gradient: while the discriminator improves, the *cooperative* generation improves as well.

Given that $D(s_{coop}) >= D(s_{gen})$, the generator will almost surely improve when trained on $s_{coop}$, for a large number of training steps, as long as the discriminator has an advantage, without requiring it to be optimal.

### A.4   Beyond A Unique Reference

In NLG, given an input, there are arguably many different possible outputs. To illustrate this, we measure the score for human written summaries compared to other gold-references: in average it obtains a ROUGE-1 of only 29.5 (std: 5.2). [4] This indicates that humans are likely to produce different sequences when given the same input. In particular, the probability to write the same exact sequence than the only gold-reference available in the training set is very low.

Should this behavior be penalized? Obviously not. And yet, this is what happens under Teacher Forcing, where, during training, any generated token that is different from the target will increase the loss. The model can therefore be exposed to contradictory information, which might limit its effectiveness. Note that this issue does not apply to a discriminator, as only two output categories (machine or human) are possible.

We argue that SelfGAN offers a theoretical solution to this multi-reference limitation. Lets denote $S_{human}$ the universe of possible correct outputs, where $s_{ref} \in S_{human}$. Then, given a perfect discriminator (optimal to distinguish real data distribution from a different distribution), and an infinite computational capacity, we have $s_{coop} \in S_{human}$. Indeed, given an infinite computational capacity, all the possible sequences can be explored. A perfect discriminator classifies a sequence $s$ as human only if $s \in S_{human}$. It results that $s_{coop} \in S_{human}$: the sequence generated via a cooperative mechanism is guaranteed to be indistinguishable from any human output, just like the reference.

In addition, since the generator probability is also taken into account in a cooperative decoding, we have $s_{coop} = argmax(P_\pi(S_{human}))$. We note that this is guaranteed only if all possible sequences are explored via an infinite computation. If we stop searching when one sequence is accepted by the decoder, it is pseudo-guaranteed since a Beam Search is only an approximation of the argmax.

$s_{coop}$ is the sequence among all the human sequences that maximise the likelihood according to the generator $\pi$. Therefore, is the generator outputs a human-level sequence (i.e. $s \in S_{human}$), is will actually correspond to $s_{coop}$. It results that considering $s_{coop}$ as the gold-reference in Teacher Forcing, the generator will not be subject to an artificial loss.

In conclusion, SelfGAN can be interpreted as a generalisation of Teacher Forcing that takes into account the multiple possible references and trains the model on the reference the highest to its likelihood.

---

[4]We used a validation set of 100 articles from the CNN/DM corpus paired with 11 different gold-references released by Fabbri et al. [9].

## A.5 Human Validation

Raters for the human validation study devoted in average 5 hours to the task and were rewarded with vouchers.