# OpenReview forum: "To Beam Or Not To Beam: That is a Question of Cooperation for Language GANs"
_NeurIPS.cc/2021/Conference — NeurIPS 2021 Poster_

### Official Review · Reviewer_iqLB · 2021-06-26

**Rating:** 6
**Confidence:** 2

**Summary:**

This work proposes selfGAN. Instead of using RL techniques to utilize the discriminator. They do Coop-MCTS which is termed as a "cooperative decoding", which is designed to generate sequences that is better than naïve sampling or decoding. And the coop-decoded sequence is then used to train the generator in turn, in the standard teacher-forcing way. I understand this approach as the process in which the discriminator is helping the generator to refine its generations. Better results comparing to MLE and ColdGAN are shown in the experiments.


**Limitations And Societal Impact:**

Did you mention how fast is MCTS comparing to beam-search?

**Main Review:**

Strength of this work: (1) The idea is novel and interesting (it reminds me of self training). (2) The results are good and the comparison protocols are principled.

Weakness: (1) There are a number of text GAN works that claim to outperform the MLE baseline, therefore only comparing with ColdGAN does not seem to be enough. (2) This paper is well written in most parts, however, I find it hard to understand the Coop-MCTS section.

In addition, I have several questions:

Line 27: "However, NLG metrics are known to not reflect well human judgement [21], which explains why the resulting models tend to be qualitatively worse than their MLE baselines [3]. " While that maybe true, it's also possible that the RL methods is simply hard to train well, and the NLG metrics is not to blame.

I did not quite understand the Coop-MCTS algorithm (I admit that I'm not familiar with PUCT). How is it able to revise a sequence? What's the inutition behind Eq.1? I hope you could rewrite this section when you revise the paper.

Is selfGAN also initialized by T5? I'm assuming it is, could the training algorithm work if it is trained from scratch?

It would be good to discuss a bit about [1], I know the two are doing different things. But the naming could be confusing.

Figure 2 left is broken?

[1] CoT: Cooperative Training for Generative Modeling of Discrete Data  https://arxiv.org/abs/1804.03782


**Time Spent Reviewing:**

2.5

---

> ### Author Response · Authors · 2021-08-09
> **Answer to: *Official Review of Paper1763 by Reviewer iqLB ***
>
> **Weakness:**
>
> Most of the language GANs are designed and compared to be effective on Unconditional Text Generation. In this work we primarily have focused on Conditional Text Generation. We therefore have chosen to compare to ColdGAN for two reasons:
> - To the best of our knowledge, ColdGAN is the only GANs that is tested on Conditional Text Generation.
> -It was published at NeurIPS 2020 and can be considered as a SOTA for GANs both for Conditional and Unconditional Generation.
> Nonetheless, we report additional results on Unconditional Text Generation in the general response. It also allows us to directly compare our approach to several other GANs including ScratchGAN, SeqGAN, MaliGAN, RankGAN, and LeakGAN. We hope that this allows addressing the concern about not comparing to other GANs.
>
> **Questions:**
>
> 1/ *Line 27: "However, NLG metrics are known to not reflect well human judgement [21], which explains why the resulting models tend to be qualitatively worse than their MLE baselines [3]. " While that may be true, it's also possible that the RL methods is simply hard to train well, and the NLG metrics is not to blame.*
>
>  - i) When reinforcement is applied on NLG models, the metric considered as reward tends to largely improve. For instance, in [23] (see Table 1 https://arxiv.org/abs/1705.04304), the ROUGE-L improves from 34.99 to 39.08 thanks to RL. This result indicates that RL methods are effective in their objective to maximize a specific reward.
>  - ii) In the same paper, the qualitative analysis shows that despite a higher ROUGE, the model that uses RL suffers from poor readability compared to the MLE baseline (see their human evaluation in Table 5).
> It seems that the RL method was actually efficient to improve its reward, i.e. ROUGE, but in such a way that the model exploits the ROUGE weaknesses and doesn't reflect human judgment well anymore.
> The authors of [23] also conclude that: “We saw that despite their common use for evaluation, ROUGE scores have their shortcomings and should not be the only metric to optimize on a summarization model for long sequences.”
>
>
> 2/ *I did not quite understand the Coop-MCTS algorithm (I admit that I'm not familiar with PUCT). How is it able to revise a sequence? What's the inutition behind Eq.1? I hope you could rewrite this section when you revise the paper.*
>
> In Equation 1,
> - $Q(s, \omega)$ (left part) corresponds to the value of selecting the action $\omega$ (i.e. the token $\omega$), in state s;
> - $U(s, \omega) = c_{\text {puct }} \pi_{\tau}(\omega \mid s) \frac{\sqrt{\sum_{b} N(s, b)}}{1+N(s, \omega)}$ (we note $U$ the right part of Eq. 1):
> 	- $\pi$ is the prior probability for token $\omega$ from our generator
> 	- $N$ is the visit count, or the number of times we’ve taken this action during current simulations:
> 		- At the numerator we sum across all the visited tokens at this state
> 		- At the denominator $N(s, \omega)$ is the number of visits for the token $\omega$. Therefore, the less we have tried this action, the greater U will be. *This encourages exploration.*
> 		- => By increasing $c_{\text {puct }}$, we put more weight toward this exploration term. By decreasing it, we more strongly value exploiting the expected result (Q). This is closely inspired by classical UCT selection scores and already used in many MCTS methods such as those considered for instance in AlphaGo.
>
>
>
> We hope it clarifies Equation 1 and gives more intuition. If so, we will rewrite the section in this direction when revising the paper.
>
>
>
>
> 3/ *Is selfGAN also initialized by T5? I'm assuming it is, could the training algorithm work if it is trained from scratch?*
>
> => SelfGAN is indeed also initialized by T5.  In the general response, our experiment on Unconditional Generation also shows that it can work from scratch.
>
> 4/ *It would be good to discuss a bit about [1], I know the two are doing different things. But the naming could be confusing.*
>
> => We thank you for suggesting to discuss their paper, it was not only relevant but also a great read, and we will discuss it in our related work.
> Their paper and our work share one similar objective, mitigating the exposure bias.
> GAN can be seen as moving from optimizing a Kullback Leibler divergence (MLE) to a Jensen-Shannon (JSD) divergence.
> In their work, the authors start from the observation that GANs on sequential discrete data are algorithms of high variance, with poor convergence (see their *Limitations of SeqGAN & its Variants* in Section 2.2).
> They proposed an approach different from a GAN to optimize a JSD via a *mediator*, which is a density function that estimates a mixture distribution of the learned generative distribution and target latent distribution.
> Therefore, the approach can be seen as a distribution estimation, offering less flexibility than a GAN framework, and not allowing guided decoding schemes. Conversely, our approach that builds on the GAN's paradigm could have applications in Controlled Text Generation (see our answer about FUDGE to reviewer sMqm).
> Note that the similarities are limited to the training part since we also proposed a new decoding mechanism which is not the case in their paper.
>
> 5/ *Figure 2 left is broken?*
>
> => Thanks for noticing this, the complete legend for the axis Y is “Discr. Gradient Magnitude”
>
> 6/ Did you mention how fast is MCTS comparing to beam-search?
>
> => An evaluation via Beam Search takes 8 minutes on CNN/DM and 5 minutes on Question Generation. An evaluation via Coop-MCTS takes 56 minutes on CNN/DM and 25 minutes on Question Generation, which is longer but compensated by the qualitative improvement the method offers.
>
> Thank you again for your effort to write a constructive review.

---

> > ### Comment · Reviewer_iqLB · 2021-08-22
> > **thanks!**
> >
> > The response is helpful, thanks!

---

### Official Review · Reviewer_sMqm · 2021-07-17

**Rating:** 6
**Confidence:** 4

**Summary:**

To train language GANs, previous approaches use the discriminator score as a reward, and train the generator through RL. Instead, this paper proposes to use the discriminator to guide the decoding process, searching for outputs with high discriminator scores, and then train the generator to generate these outputs. To find outputs with high discriminator scores, the authors employ Monte Carlo Tree Search and use the discriminator as the value network. Experiments include question generation and summarization, and the proposed method shows improvements to MLE and ColdGAN.

**Limitations And Societal Impact:**

The authors have discussed the societal impact of natural language generation in the paper.

**Main Review:**

While the proposed model looks fine, I think the experiments and baselines are lacking, and most importantly, the results do not match those reported in previous work. I hope the authors can validly address my concerns.

The results of MLE and ColdGAN in Table 1 are overall much worse than those reported in the ColdGAN paper. Just to give an example, T5-small (MLE) has BLUE-4 19.65 on question generation (see Table 2 in the ColdGAN paper), while this paper reports 16.5. Can the authors please explain the huge discrepancy?

In addition, the models are not tested on unconditional generation. We know that mode collapse is a serious problem of GANs. It's thus unclear whether the proposed method can generate diverse outputs in unconditional scenarios.

For cooperative decoding, the proposed MCTS approach is compared against a reranking baseline. But there're also other methods to use the classifier to guide the generation process. For example, one can combine the generator probabilities and the discriminator probabilities to sample the next word [1]. Can the authors compare with this method as well?

In Fig 2 Right, the authors report the collinearity of generator gradients for generated samples with those for human references. However, it's unclear to me why the higher the better. If so, why not just train with human references using MLE, which has a collinearity of 1?

Finally, the writing is sometimes obscure (e.g., "the sampling mechanism of sequences that have to be favored through MLE", "for each condition in the training set X"). Please revise them to avoid confusion.

[1] Yang and Klein. FUDGE: Controlled Text Generation With Future Discriminators. NAACL 2021.

**Time Spent Reviewing:**

5h

---

> ### Author Response · Authors · 2021-08-09
> **Answer to: *Official Review of Paper1763 by Reviewer sMqm ***
>
> We thank the Reviewer for the detailed review. We hope to validly address your concerns in our answer, please let us know if we did not fully address them. We believe that it will definitely contribute to improving the current version of the paper.
>
> i) We have tested the model on Unconditional Text Generation and report the results in the general response.
>
> ii) About the discrepancy of results between the ColdGAN paper and our Table 1:
> We noticed this discrepancy and have investigated it: the root cause is the difference in terms of hyperparameters when finetuning the MLE baseline. In particular, ColdGAN authors finetuned their MLE baseline during 20 epochs and a batch size of 64. We did set the number of epochs to 5 and the batch size to 12 for our MLE baseline, which performs worst.
> We had already contacted the ColdGAN’s authors who shared with us their MLE checkpoints. Starting from these better MLE baselines, all our results have improved. We now obtain consistent results with ColdGAN. Importantly, the relative differences between the models all remain consistent with our Table 1. In particular, here are a selected sample of the results when initialised with the better checkpoint:
>
> |             		|    |           	         | QG     |         |             |           |             |  \|  | Sum.  |     | 	      | 	 |            |
> |------------------------|-------------------|---|--------|--------|----------|----------|-----------|---|-------|-------|---------------|--------|----------|
> |  Generator          |  Decoder	        | \| | B4     | R1     | RL     | Base  | Base+ | \| |    B4   | R1     | RL       | Base | Base+ |
> |          MLE           | BeamSearch | \| | 19.65 | 45.23 | 41.12 | 15.3% | 15.1% | \|  | 15.94 | 42.34 | 40.37 | 08.8% | 08.1% |
> |          MLE           | MCTS            | \| |19.82 | 45.31 | 41.55 | 33.3% | 20.6% | \|  | 16.31 | 42.51 | 40.61 | 20.1% | 12.1% |
> |          ColdGAN   | BeamSearch | \| |19.93 | 45.23 | 41.38 | 25.8% | 17.9% | \|  | 16.29 | 42.84 | 40.76 | 14.7% | 10.5% |
> | SelfGAN_MCTS | BeamSearch | \| | 20.48 | 46.98 | 42.63 | 33.9% | 21.6% | \|  | 16.99 | 42.80 | 41.48 | 20.5% | 12.8% |
> | SelfGAN_MCTS | MCTS	       | \| | 21.12 | 48.89 | 44.69 | 39.9% | 26.2% | \|  | 17.58 | 43.52 | 42.26 | 23.3% | 15.3% |
>
> - Decoding with MCTS systematically performs favorably compared to BeamSearch
> - SelfGAN improves over MLE and ColdGAN
> This additional result also indicates that other checkpoints can benefit from SelfGAN without the need to tune any hyperparameters, which is positive in terms of robustness/stability.
>
> iii) About FUDGE:
> *“For cooperative decoding, the proposed MCTS approach is compared against a reranking baseline. But there are also other methods to use the classifier to guide the generation process. For example, one can combine the generator probabilities and the discriminator probabilities to sample the next word [1]. Can the authors compare with this method as well?”*
>
> => Actually the FUDGE method can be seen as a reranking method as mentioned by the authors, see their Section 3: *Intuitively, one can view $\beta$   as rescoring or reranking $\gamma$’s original hypotheses.*
> Their formulation as detailed in their equations and depicted in their Figure 1 corresponds exactly to the $DAS_{local}$ paper. The main difference is that $DAS_{local}$ uses a single discriminator aiming to learn high level features, while FUDGE extends it to N discriminators that are specialized (e.g.one classifier is trained to predict if the generated text has a rhythm for poetry generation).
> In the context of our paper where we have only one discriminator, the differences between DAS and FUDGE are therefore limited to the model choice and its hyperparameters:
> - DAS relies on T5 while FUDGE chose an LSTM;
> - hyperparameters: e.g. FUDGE allows a discriminator’s reranking of the 200th best tokens given by the generator. DAS limits it to 20 (which is ten times faster).
>
> Following your suggestion, we have conducted an additional experiment to compare FUDGE with the other approaches: $MLE + FUDGE$, where we use the FUDGE hyperparameters instead of the DAS ones. For a fair comparison, we still rely on T5 for the discriminator (T5 is expected to perform better than a randomly initialized LSTM). It performs slightly better than $DAS_{global}$ but worse than $DAS_{local}$, and worse than $Coop-MCTS$.
> This good performance of our $Coop-MCTS$ compared to $FUDGE$ indicates that our decoding method could also have applications in the field of Controlled Text Generation, not only GANs.
> Finally, we would like to emphasize that cooperative decoding is orthogonal to our proposed SelfGAN framework. Future works that will improve over our Coop-MCTS will therefore directly benefit our proposed framework.
>
> iv) *In Fig 2 Right, the authors report the collinearity of generator gradients for generated samples with those for human references. However, it's unclear to me why the higher the better. If so, why not just train with human references using MLE, which has a collinearity of 1?*
>
> A specificity in NLG is that several texts can be correct, not only the single reference available.  Also, using only ground truth references suffers from the well known exposition bias, which is not the case when considering sampled texts from the generator during training. However, we would like to use texts that sound like progressively more resemblant to the references. The fact that generated texts from SelfGAN become more and more collinear with the reference (in terms of generator gradients) indicates that they actually impact training in the desirable direction. Beyond the teacher forcing limitations where the training is restricted to see ground truth references only, this better collinearity shows that our process progressively uses samples that have a positive impact on training, converging to the generation of more and more realistic samples.
>
> Finally, thank you for pointing out these confusing sentences. If accepted, we will clarify them in the final version.

---

> > ### Comment · Reviewer_sMqm · 2021-08-21
> > **Thanks for your reply**
> >
> > I increased my score to 6, as the authors resolved the discrepancy with the results in the ColdGAN paper and added an unconditional generation experiment.
> >
> > About FUDGE, it combines the generator probabilities **and** the discriminator probabilities. You said it's equivalent to DAS, but I thought DAS reranks **solely** based on the discriminator probabilities?

---

> > > ### Author Response · Authors · 2021-08-23
> > > **Thanks and about FUDGE Vs DAS**
> > >
> > > Thanks for letting us know!
> > >
> > > About FUDGE Vs DAS:
> > > DAS also combines both, see Equation 4 in their paper (http://proceedings.mlr.press/v119/scialom20a/scialom20a.pdf) and the hyperparameter $\alpha$. The authors discussed the choice for $\alpha$ at the end of their preliminary study and then have chosen $\alpha=1$, which we, therefore, used as well. We will include their Eq. 4 when describing DAS, which should help to clarify.

---

### Official Review · Reviewer_TEo2 · 2021-07-19

**Rating:** 7
**Confidence:** 3

**Summary:**

This paper presents a method of training (or rather fine-tuning) text generation models using a GAN-like setup. Instead of using the discriminator directly to update the generator as is usually the case, the authors propose using the discriminator as a way of cooperative decoding to generate high-quality samples which are then used as the gold output to train the model in the usual teacher-forcing setup. The authors explore a sampling procedure based on MCTS to generate more diverse sentences rather than using beam-search or sampling during training and inference. On summarization and question generation, they show improvements on the standard and proposed evaluation metrics.

**Limitations And Societal Impact:**

Yes

**Main Review:**

Strengths:
1. Language GANs have been shown to be difficult to train due to sparse signals from the discriminator. This paper presents an interesting idea to incorporate this signal in a more stable manner.
2. MCTS based decoding algorithm also has potential applications even for cases where the proposed training method was not used.
3. The paper is clearly written and easy to follow for the most part.

Weaknesses and comments:
1. The paper presents the method as a language GAN yet doesn't evaluate it on the most common task used in this area: unconditional text generation where the goal is to generate high-quality diverse text where the benefits of the proposed decoding method can be more clearly evaluated.
2. Relating to my previous comment, a diversity measure should be added to the evaluation. Plus the quality metrics show very small improvements in most cases over baselines.
3.  The proposed decoding process is quite slow for the improvement it offers.


------------------------

Update: The author response addresses most of my concerns, I am updating my score to 7.

**Time Spent Reviewing:**

4

---

> ### Author Response · Authors · 2021-08-09
> **Answer to *Review of Paper1763 by Reviewer TEo2***
>
> We thank the Reviewer for the insightful comments.
>
> **About the *small* improvement offered by our model:**
>
> => The improvement in ROUGE & BLEU is limited to ~one point. However, those metrics are known to not reflect well human judgement (see our citation [21]).
>
> Beyond ROUGE & BLEU, we would like to emphasize that the improvement is not that small between our best model ($SelfGAN_{Coop-MCTS} + Coop-MCTS$ ), compared to the best previous work ($ColdGan+BeamSearch$), on both:
> - human evaluation (see Table 2): we obtain a relative improvement of 15% for coherence (3.3 to 3.9) and relevance (3.5 to 4.2);
> - the discriminator-based metrics (see Table 1): we obtain a relative improvement over 20% (20.6 to 26.2 on Question Generation and 11.8 to 15.3 on Summarization).
>
> We show that SelfGAN significantly improves quality on more than 15% of generated texts w.r.t. competing systems: SelfGAN has 15% more of its outputs that are classified as human-like, compared to ColdGAN or DAS. We argue that such improvements can have important benefits for downstream applications.
>
> **About Unconditional Generation and a diversity measure:**
> We address the two mentioned weaknesses in the general answer by reporting the results for Unconditional Generation, including the diversity (via Self-BLEU).
>
> We hope our answer properly addresses all your concerns and we thank you again for the insightful review.

---

> > ### Comment · Reviewer_TEo2 · 2021-08-31
> > **Thank you for the clarifications**
> >
> > I would like to thank the authors for their detailed response including the results for unconditional generation. Please include them in your final draft. I will increase my score to 7.

---

### Official Review · Reviewer_oytt · 2021-07-19

**Rating:** 7
**Confidence:** 4

**Summary:**

This paper attempts to address the difficulties of adversarial training for language generation, as discrete generator outputs make conventional GANs difficult to train.

The authors propose combining cooperative decoding proposed by Gabriel et al (2019), along with MCTS-based decoding, to train a discriminator and generator jointly. The generator outputs are trained with the results of their MCTS-based cooperative decoding procedure.

The authors demonstrate their method's performance on summarization (CNN and Daily Mail dataset) and question generation (SQuAD).


**Limitations And Societal Impact:**

The authors adequately discuss limitations and societal impact in their conclusions.

**Main Review:**

Overall, this work demonstrates a novel approach to integrating cooperative training, and is a useful contribution. The method is promising, and could be extended and applied to other discrete generation applications as well.

The comments that follow are mostly related to experiments and the paper's presentation.

Regarding experiments, as the authors point out, the "Base" discriminator metric uses the discriminator trained on MLE outputs generated by beam search. However, it's used both as a metric and for training, so the authors use a "Base+" discriminator that is further fine tuned on the final model outputs. One potential concern is that the fine tuning process may result in the finetuned model still being too similar to Base, and unable to find discriminating factors that would be found had the model been trained from scratch. Have the authors considered training a discriminator from scratch, and if so, how does it perform? Performing this experiment is not a requirement for my review: as generated natural language is difficult to evaluate automatically, human evaluations should still take precedence.

To clarify, is the generator pretrained with standard MLE before training SelfGAN/Coop-MCTS, or is the latter trained from scratch? Or is only the discriminator pretrained? How stable is the SelfGAN training?

Overall, the paper may be somewhat clearer with more equations to precisely and concisely describe the method, and/or some minor reorganization. Algorithm 1 is somewhat brief, and the overall method's description is split between Section 3 and the MCTS description in Section 4, which due to the author's setup, are also important for describing how the generator is trained.

*Related work:*

The authors cite the cooperative decoding work of Gabriel et al from 2019, but it appears they have a later version in EACL 2021, see https://arxiv.org/abs/1907.01272. This should be updated accordingly.

As the authors cite other applications of MCTS in NLP, the authors should also consider citing related MCTS decoding work in neural machine translation, such as https://arxiv.org/abs/2004.12527 and https://arxiv.org/abs/2104.05336.

*Clarifications:*

- On line 309, the sentence "SelfGAN produces sequences that form a more compact set for discriminator training …" is unclear, what does this refer to?
- On line 319, can you clarify how the metric for "colinearity of generator gradients for the generated samples from the model with those for the corresponding human references" is computed?
- Last sentence of the conclusion is awkward/unclear, what does "increasing it" refer to?

*Minor comments:*

- L111: add a caption for Algorithm 1, and reference it in the main text
- L124 and alg. 1 L5: inconsistent naming between "standard training" and "classical maximization", perhaps say something like "standard maximum likelihood with S_coop as the target"
- L124: in the first equation, shouldn't $\sum_{...} \pi_{\theta}(...)$ be $\sum_{...} \log \pi_{\theta}(...)$?
- L178: 'a in' -> 'in'
- L262: version -> versions
- Table 1 should explain "B4", "R1", and "RL" in the caption (e.g. BLEU4 ("B4"))
- L323: mean -> means?


**Time Spent Reviewing:**

4

---

> ### Author Response · Authors · 2021-08-09
> **Answer to  *Official Review of Paper1763 by Reviewer oytt***
>
> We thank the Reviewer for his detailed review.
>
> **Main Review:**
>
> *Is the generator pretrained with standard MLE before training SelfGAN/Coop-MCTS, or is the latter trained from scratch? Or is only the discriminator pretrained?*
>
> => In our experiments, both are pretrained using the T5-small weights. We discuss an additional experiment in the general response where both were randomly initialized instead, i.e SelfGAN from scratch.
>
> About BASE/BASE+: *Have the authors considered training a discriminator from scratch, and if so, how does it perform?*
>
> => We have tested $BASE+_{random}$ corresponding to $BASE+$ but this time initialised randomly (with XAVIER):
> 1) In absolute we obtain higher scores than $BASE+$ (e.g. in Table 1 for QG/Sum, MLE: 22.1/15.3; for $SelfGAN_{Coop-MCTS} + Coop-MCTS$: 35.4/25.8). This means that as expected, not pretraining our discriminator reduces its ability to distinguish between human and machine-generated texts.
> 2) Comparatively, the differences between the different models remain similar and importantly the order between the models is preserved.
>
>
> **Clarification:**
>
> *On line 309, the sentence "SelfGAN produces sequences that form a more compact set for discriminator training …" is unclear, what does this refer to?*
>
> => One of the identified causes of instability for a GAN’s discriminator is the sudden generation of unlikely sequences, for which the discriminator is not accurate at determining if they sound closer to human or machine. Such unlikely sequences are prone to induce important instabilities for its discriminator over time, which can be indicated by highly fluctuating gradient magnitude.
> Conversely, by a more compact set of sequences, we mean that SelfGAN produces more stable sequences over time.
>
> *On line 319, can you clarify how the metric for "colinearity of generator gradients for the generated samples from the model with those for the corresponding human references" is computed?*
>
>  => We obtain the two gradient vectors among all the parameters, using two different *references*: i) using the human references, and ii) using the generated sample from the corresponding model. We report the average dot product of such vectors, which we will clarify in the text.
>
> *The last sentence of the conclusion is awkward/unclear, what does "increasing it" refer to?*
>
> => increasing the stability of the discriminator
>
> Thank you for pointing out the minor comments as well as the missing citations and the last version of Gabriel et al in EACL 2021. We have now fixed and completed them accordingly.

---

### Author Response · Authors · 2021-08-09
**General answer**

We would like to thank the reviewers for the time and effort they have dedicated to provide their valuable feedback, and we believe it will improve the paper. We look forward to hearing from the reviewers and responding to any further questions and comments they may have.

One important concern from Reviewers **TEo2** and **sMqm** is the absence of experiments on **Unconditional Text Generation**. In the following, we address this concern by reporting the results we had on Unconditional Text Generation. This also allows to address the concern of Reviewer **iqLB** that we only compared to ColdGAN: the following results allow us to compare our GAN to several other previous works. Finally, the results also contain a comparison with a SelfGAN trained from scratch.

As a preamble, we would like to detail why we chose to focus only on Conditional Text Generation in the submitted version:
- i) those are downstream tasks with real-world applications, while Unconditional Text Generation direct applications are less clear;
- ii) those two downstream tasks are very popular in general, not only for comparing GANs. We believe this makes it even more challenging to improve over MLE.
Nonetheless, our results on Unconditional Text Generation are also positive. If accepted, we will use the additional page to report them.

**Experimental setup:** we followed the ColdGAN setup:
We compared our proposed approaches on the EMNLP 2017 News dataset.
The evaluation takes into account both the quality and the diversity. Consistently with previous works (e.g. ColdGAN, ScratchGAN, LeakGAN), we use the following metrics:
-  BLEU-5 for measuring the quality (higher better);
- Self-BLEU-5  for measuring the diversity (lower better).

To obtain a finer comparison between models, Caccia et al. [3] proposed to draw the curve of BLEU vs self-BLEU, by sampling with various temperatures at inference.

We denote as $Sample$ the standard method to generate text in this setup, as used in all the previous works we are comparing to. It is a simple left to right decoding where, at each step, a token is sampled among the Softmax probabilities scaled by the temperature.

In our *Coop-MCTS*, the probability of a token is given by its visit counts during the simulations. In Conditional generation, we select at each step the token with the maximum number of counts as detailed in Line 195. In Unconditional generation, we sample from the tokens counts distribution.

**Results:**
For the MLE and ColdGAN models, we asked the ColdGAN authors to share their checkpoints: this allows us to directly compare our proposed methods w.r.t. ColdGAN,  ScratchGAN, SeqGAN, MaliGAN, RankGAN, and LeakGAN as reported in Figure 2 of the ColdGAN paper (https://arxiv.org/pdf/2006.04643.pdf).

Results on Unconditional Text Generation for samples realized at three different temperatures, in terms of BLEU;Self-BLEU (higher better;lower better):

| Model                | T=0.5 | T=1 | T=2 |
|-----------------------------------|-----------|-----------|-----------|
| $MLE + Sample$                | 0.42;0.29 | 0.31;0.11 | 0.18;0.07 |
| $ColdGAN + Sample$            | 0.47;0.21 | 0.33;0.08  | 0.22;0.06 |
| $MLE +  Coop-MCTS$                      | 0.45;0.22 | 0.34;0.10 | 0.21;0.06 |
| $SelfGAN_{Coop-MCTS} + Coop-MCTS$ | 0.48;0.20 | 0.37;0.09  | 0.24;0.05 |


Overall, the results are consistent with the experiments on Conditional Generation:
- The MLE generator decoded with our proposed MCTS (3rd row) obtains:
  - significantly better results than when the same MLE decoded with Sample(1st row);
  - slightly lower results than ColdGAN decoded with Sample(2nd row).
- SelfGAN decoded with MCTS (4th row) obtains the best results.

Finally, we experimented with SelfGAN from scratch. In the following Table, all the models were randomly initialized:

| Model                | T=0.5 | T=1 | T=2 |
|-----------------------------------|-----------|-----------|-----------|
| $MLE + Sample$                | 0.33;0.4 | 0.27;0.3 | 0.21;0.21 |
| $ScratchGAN + Sample$            | 0.31;0.42 | 0.25;0.29  | 0.19;0.22 |
| $SelfGAN_{Coop-MCTS} + Sample$ | 0.33;0.39 | 0.25;0.27  | 0.21;0.19 |
| $SelfGAN_{Coop-MCTS} + Coop-MCTS$ | 0.36;0.40 | 0.27;0.25  | 0.20;0.15 |

Our proposed SelfGAN from scratch compares favorably to MLE from scratch and ScratchGAN. Decoded with MCTS improves even more, obtaining the best results among models that were not pretrained.

ScratchGAN is one of the best GAN to our knowledge; to obtain more comparisons see Figure 2 in https://arxiv.org/pdf/2006.04643.pdf.

As mentioned by Reviewers TEo2 and sMqm, this experiment on Unconditional Text Generation allows us to show that our proposed models generate high-quality diverse text and don’t suffer from mode collapse. If accepted, we will include these results using the additional page available.

We thank you again for your time and efforts.

---

### Decision · Program_Chairs · 2021-09-27

**Decision:**

Accept (Poster)

**Comment:**

This work demonstrates the effectiveness of a novel and interesting approach to utilising GANs with cooperative training and MCTS. The paper is well written and easy to follow. Though the initial results in the paper lacked evaluation on an important task (unconditional text generation) and didn't evaluate the diversity of generated text, the strong rebuttal resolves these issues and gives convincing evidence of the effectiveness of SelfGAN and MCTS. Hopefully this paper inspires more work in this interesting direction.